# Cross-sectional analysis of national testosterone prescribing through prescription drug monitoring programs, 2018–2022

**Scott Selinger**[1]☯*, **Aneesh Thallapureddy**[2]☯

**1** Department of Internal Medicine, Division of General Internal Medicine, University of Texas at Austin Dell Medical School, Austin, Texas, United States of America, **2** University of Texas at Austin Dell Medical School, Austin, Texas, United States of America

☯ These authors contributed equally to this work.

* scott.selinger@austin.utexas.edu

**Data Availability Statement:** Data cannot be shared publicly because all of the data was obtained through separate data use agreements

## Abstract

### Background

For two decades preceding the COVID-19 pandemic, testosterone therapy (TT) became more prevalent in the US. Given the forced shift in practice patterns and healthcare accessibility during the pandemic, it was unclear how TT utilization would change.

### Objective

To assess the change in testosterone prescriptions nationally.

### Design

Cross-sectional study.

### Data sources

State prescription drug monitoring program data between 2018 and 2022.

### Participants

All individuals filling testosterone prescriptions in participating states.

### Measurements

Unique people filling testosterone prescriptions annually, demographic information on gender and age as available.

### Results

In 2022 there was a 27% relative increase of subjects treated with TT (+439,659 cases compared with 2018). The increase was more evident in the pandemic period with a rise in prevalence most notable for people 45–54 (114,114 people, 35% increase) and 35–44 (97,263 people, 58% increase). All regions except the Midwest increased the total population

with individual states participating in this study. Some of those agreements obligate us to destroy the data after publication, others prohibit the data from being shared without permission from the governing body, and some data does not have that restriction. All of the data obtained was freely available on individual request from the following institutional points of contact, and data may be shared by these institutions on reasonable request and with the completion of a Data Use Agreement. Requests may be directed to the institutions listed in the Supplementary file titled 'Contact information for participating states.

**Funding:** The author(s) received no specific funding for this work.

**Competing interests:** The authors have declared that no competing interests exist.

treated, led by the South (52%) followed by the West (28%) and Northeast (23%). Available data indicated men accounted for most patients treated in all age groups except under 24 years.

## Limitations

Study population limited to those in participating states with no diagnostic information and limited demographics available.

## Conclusion

Between 2018 and 2022, and primarily after the start of the pandemic in 2020, nationally there was a substantial increase in the number of people using TT. The largest increases occurred in a younger demographic, primarily men, than have previously been reported or studied. These results echo other findings showing increased use of controlled substances during the pandemic period and warrant further study regarding the factors behind this rise.

## Introduction

Testosterone therapy (TT) prescribing has experienced significant growth since 2000 [1]. The number of new and total users grew for the first decade of this millennium before decreasing in 2013 when research signaled an increased risk of myocardial infarction and stroke associated with testosterone use [2]. Direct marketing of TT towards men has increased [3] and our earlier analysis of the Texas Prescription Drug Monitoring Program (PDMP) revealed a recent and substantial rise in the population-level incidence of TT [4], strongly driven by new treatment of a younger population than has typically been included in prior research. The goal of this study was to assess the changes in prescribing patterns nationally to see if this reflected a larger trend.

## Materials & methods

This cross-sectional study was exempted from review by The University of Texas at Austin Dell Medical School institutional review board. Patient informed consent was waived owing to the use of deidentified and aggregated data. This study followed the Strengthening the Reporting of Observational Studies in Epidemiology (STROBE) reporting guidelines.

We sought prescription data through PDMPs for fifty states and the District of Columbia. With their cataloging of controlled substance prescriptions filled at state pharmacies, data was requested containing demographic characteristics of all people filling prescriptions for controlled substances at outpatient pharmacies from all fifty states and the District of Colombia. Data was requested and analyzed between April and December 2023.

To assemble the cohort, state PDMPs or their supervising authority were contacted individually for participation. Due to administrative and regulatory hurdles varying by state, our data request refocused from deidentified prescription records to aggregated numbers of unique patients treated with testosterone annually, broken down for six patient age groups (<24, 25–34, 35–44, 45–54, 55–64, and 65+). National Drug Codes and drug names were used to identify testosterone prescriptions. Diagnosis codes and treatment indications were not available and gender information was not consistently available.

To account for population growth and interstate migration, population estimates were obtained from the United States Census between 2018 and 2022 to evaluate the population percentage receiving a prescription for TT. Prevalence rates for the populations for our defined age groups were also calculated. To consider changes during the COVID-19 pandemic, we compared a linear forecast of patients treated with TT based on 2018–2020 data with the actual 2021–2022 values and visualized the difference in prevalence percentage using Microsoft Excel for Microsoft 365 MSO (Version 2310).

## Results

Of the 51 entities contacted, 25 were included for analysis. Reasons for non-participation were legislative prohibition of data-sharing, lack of data collection during the defined period and lack of response to multiple data requests, delineated by state in Table 1. Participating entities controlling PDMP data are listed in Table 2. Most states performed the analyses and shared the aggregated data while Arkansas, Iowa, Kentucky, New Jersey, Texas, and Wisconsin provided access to deidentified data for our analysis. Due to data classification incongruent with our age stratification, Wisconsin was excluded from the final analysis.

Within the study period, the number of people in the total sample treated with testosterone annually increased 27% from 1,216,982 to 1,656,641 in 2022. Considered regionally, this increase was largest in the South at 52% (AL, AR, GA, KY, LA, OK, SC, TN, TX, VA), followed by the West at 28% (AZ, CO, OR, UT, WA) and Northeast at 23% (MD, MA, NH, NJ, PA, RI), while a decrease of 7% was seen in the Midwest (IA, KS, OH). Considering population changes, the percentage of the study population filling a testosterone prescription increased 30% from 0.76% to 0.98%. There was similarly varied regional growth seen in Fig 1 with TT prevalence decreasing in the Midwest (-24%) and rising in the Northeast (20%), West (24%) and South (47%).

Age group analysis (Table 3) shows increases in TT prevalence seen in those 24 and under (120%), 25–34 (86%), 35–44 (45%), 45–54 (35%), 55–64 (17%) and 65 and older (12%). The largest absolute increases were seen in the middle age groups, 45–54 (114,114) and 35–44 (97,263). Compared to the overall study period, the increase in TT patients between 2020–2022 accounted for most of the growth in all age groups (Fig 2)– 24 and under (51%), 25–34 (69%), 35–44 (74%), 45–54 (83%), 55–64 (93%), 65 and older (85%). The interstate differences in actual vs expected growth by population percentage in the study period are shown in Fig 3. Calculated incidence rates per 100,000 of the total population (Fig 4) showed the largest increase in the 45–54 group, followed by the 35–44 and 55–64 groups, with near convergence of the 35–44 and >65 groups.

**Table 1. Reasons provided by PDMP for non-participation in data-sharing.**

| Legislative Prohibition | Requested Data Unavailable | Did Not Respond |
|:---:|:---:|:---:|
| California | Alaska | Indiana |
| Delaware | Hawaii | Mississippi |
| District of Colombia | Illinois | North Carolina |
| Florida | Maine | South Dakota |
| Michigan | Missouri | Vermont |
| Nebraska | Montana | West Virginia |
| Nevada | New Mexico | |
| New York | Wyoming | |
| | North Dakota | |

**Table 2. Participating Entities providing PMP data.**

| |
|---|
| Alabama Department of Public Health |
| Arizona Board of Pharmacy |
| Arkansas Department of Health |
| Colorado State Board of Pharmacy |
| Georgia Department of Public Health |
| Idaho Division of Occupational & Professional Licenses |
| Iowa Department of Health & Human Services |
| Kansas Board of Pharmacy |
| Kentucky Cabinet for Health & Family Services, OIG |
| Louisiana Board of Pharmacy |
| Maryland Department of Health |
| Massachusetts Department of Public Health |
| New Hampshire Department of Health and Human Services |
| New Jersey Division of Consumer Affairs |
| Ohio Board of Pharmacy |
| Oklahoma Bureau of Narcotics and Dangerous Drugs Control |
| Oregon Health Authority |
| Pennsylvania Department of Health |
| Rhode Island Department of Health |
| South Carolina Department of Health & Environmental Control |
| Tennessee Department of Health |
| Texas State Board of Pharmacy |
| Division of Occupational and Professional Licensing (DOPL) |
| Utah Controlled Substance Database |
| Virginia Department of Health Professions |
| Washington State Department of Health |

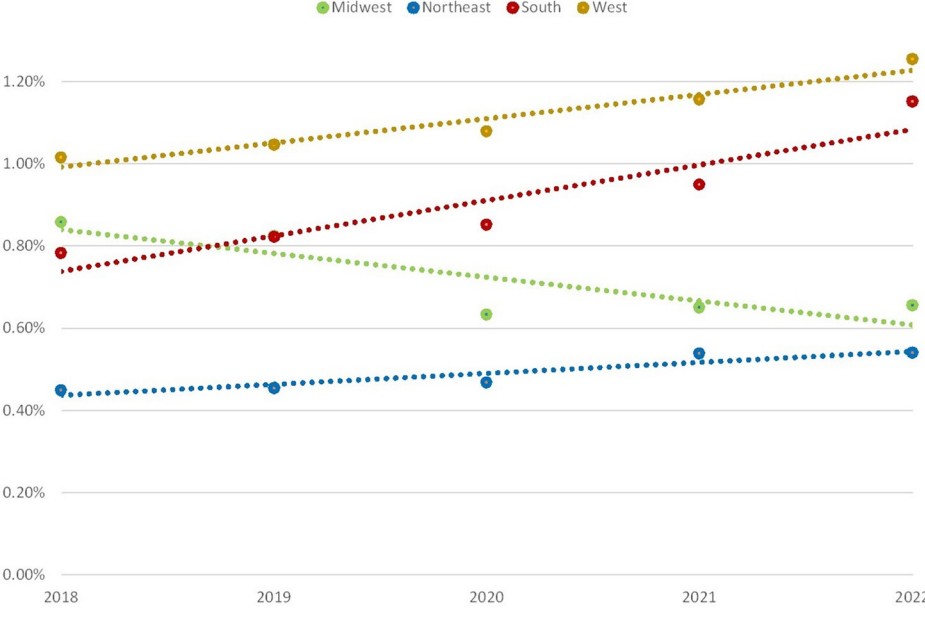

**Fig 1. Percentage of regional population filling prescription for testosterone by year.** Green—Midwest (IA, KS, OH), Blue—Northeast (MD, MA, NH, NJ, PA, RI), Red—South (AL, AR, GA, KY, LA, OK, SC, TN, TX, VA), Yellow —West (AZ, CO, OR, UT, WA).

**Table 3. Patients treated (population percentage) with testosterone therapy.**

|  |  | Midwest | Northeast | South | West | Total |
|---|---|---|---|---|---|---|
| 24 and under | 2018 | 887 (0.04) | 7839 (0.07) | 10940 (0.04) | 8190 (0.09) | 27856 (0.05) |
|  | 2019 | 1754 (0.05) | 9378 (0.08) | 13962 (0.05) | 10258 (0.11) | 35352 (0.07) |
|  | 2020 | 1937 (0.06) | 10511 (0.09) | 18902 (0.06) | 12044 (0.13) | 43394 (0.08) |
|  | 2021 | 2554 (0.08) | 12926 (0.11) | 18623 (0.06) | 14803 (0.15) | 48906 (0.09) |
|  | 2022 | 3105 (0.09) | 14822 (0.13) | 24119 (0.08) | 17654 (0.18) | 59700 (0.11) |
| 25–34 | 2018 | 5857 (0.31) | 9360 (0.19) | 26126 (0.23) | 15351 (0.36) | 56694 (0.25) |
|  | 2019 | 7767 (0.33) | 10390 (0.21) | 29347 (0.26) | 17019 (0.39) | 64523 (0.28) |
|  | 2020 | 6588 (0.29) | 12040 (0.24) | 34502 (0.31) | 19468 (0.45) | 72598 (0.32) |
|  | 2021 | 7884 (0.34) | 15217 (0.3) | 40367 (0.36) | 23509 (0.54) | 86977 (0.38) |
|  | 2022 | 8794 (0.38) | 16980 (0.34) | 54803 (0.48) | 28199 (0.64) | 108776 (0.47) |
| 35–44 | 2018 | 16061 (0.92) | 18288 (0.41) | 92989 (0.9) | 40882 (1.06) | 168220 (0.82) |
|  | 2019 | 19103 (0.89) | 18860 (0.41) | 101343 (0.97) | 43167 (1.09) | 182473 (0.87) |
|  | 2020 | 15015 (0.68) | 21055 (0.44) | 110965 (1.04) | 46289 (1.15) | 193324 (0.89) |
|  | 2021 | 16519 (0.74) | 26061 (0.54) | 126231 (1.17) | 51943 (1.26) | 220754 (1.01) |
|  | 2022 | 16548 (0.74) | 27923 (0.57) | 161129 (1.47) | 59883 (1.44) | 265483 (1.19) |
| 45–54 | 2018 | 21614 (2.72) | 39865 (0.81) | 188463 (1.68) | 74046 (2.1) | 323988 (1.58) |
|  | 2019 | 26163 (2.29) | 38916 (0.81) | 197383 (1.78) | 76544 (2.18) | 339006 (1.65) |
|  | 2020 | 24868 (2.17) | 40238 (0.83) | 198824 (1.79) | 78946 (2.23) | 342876 (1.66) |
|  | 2021 | 27303 (2.4) | 45604 (0.96) | 217302 (1.96) | 85226 (2.39) | 375435 (1.83) |
|  | 2022 | 29627 (2.59) | 45861 (0.98) | 268142 (2.42) | 94472 (2.63) | 438102 (2.14) |
| 55–64 | 2018 | 26216 (3.03) | 50860 (0.99) | 199012 (1.8) | 89411 (2.49) | 365499 (1.77) |
|  | 2019 | 30720 (2.4) | 50283 (0.98) | 206092(1.85) | 91066 (2.52) | 378161 (1.79) |
|  | 2020 | 26367 (2.07) | 51853 (0.98) | 201628 (1.8) | 90820 (2.52) | 370688 (1.74) |
|  | 2021 | 27139 (2.16) | 57006 (1.09) | 215677 (1.93) | 93471 (2.61) | 393293 (1.85) |
|  | 2022 | 28393 (2.31) | 54447 (1.05) | 255840 (2.3) | 97323 (2.75) | 436003 (2.07) |
| 65 and older | 2018 | 27878 (1.14) | 40195 (0.64) | 136474 (1.16) | 70868 (1.54) | 275415 (1.1) |
|  | 2019 | 33063 (1.08) | 40733 (0.63) | 143271 (1.18) | 73749 (1.54) | 290816 (1.1) |
|  | 2020 | 25035 (0.8) | 42058 (0.64) | 143044 (1.16) | 76370 (1.58) | 286507 (1.06) |
|  | 2021 | 24230 (0.76) | 47528 (0.7) | 158144 (1.25) | 80432 (1.61) | 310334 (1.14) |
|  | 2022 | 24981 (0.77) | 44786 (0.64) | 194632 (1.49) | 84178 (1.63) | 348577 (1.23) |

Analysis of gender-identifying information was only available for Arizona, Arkansas, Iowa, Kentucky, and New Jersey. Between 2018–2022, in all age groups, the male percentage of TT patients decreased, notably in the under 24 group where females became the majority, likely reflecting a rise in gender-affirming care. For all other age groups, men accounted for at least two-thirds of the TT population (Fig 5).

## Discussion

In this cross-sectional study, there was a substantial increase in the number of people receiving TT nationally between 2018–2022, most of which occurred after the onset of the COVID-19 pandemic in 2020. The demographics driving this increase were younger men, more prominently in the South and West than the Northeast, with diminishing usage in the Midwest, consistent with prior geographic differences in prescribing patterns [1, 5]. By 2022, TT prevalence increased in the 35–44 group to match those 65 and older.

Some of our findings are consistent with prior trends in the US. Between 2016–2019, claim numbers for TT increased nearly 50%, largely driven by an increasing proportion of

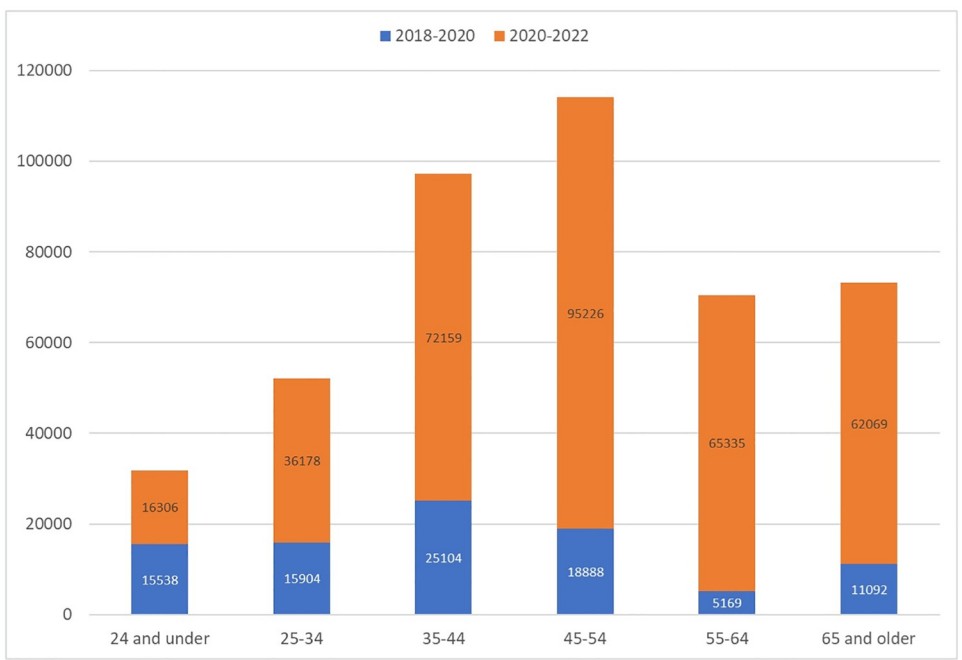

**Fig 2. Increase in population treated with testosterone in study period by age group.** Blue portion represents increase in prevalence from 2018–2020, Orange portion represents 2020–2022 period.

prescribers from primary care, internal medicine, and family practice physicians as well as nurse practitioners and physician's assistants, far more than from urologists and endocrinologists [6]. The prevalence of hypogonadism based on claims data had increased nearly 600% between 2008–2017 with similar variations in distribution geographically to our study, but with growth driven more by those older than 45 [7]. In the past, TT has been administered much more readily [8] and with more liberal interpretations for treatment initiation in the U.S [9]. than most other countries who track this data, though we could find no more recent TT analyses of prescribing patterns internationally during the COVID-19 pandemic.

The onset of the pandemic, and the societal shifts and socioeconomic pressures around it, promoted or deepened common somatic symptoms such as fatigue, poor sleep, depression, anxiety, and unintentional weight gain, all of which overlap with the non-specific symptoms of hypogonadism [10–12]. The peri-pandemic period saw increased use of mood-altering controlled substances such as opioids [13], stimulants and non-stimulant ADHD treatments, antidepressants, benzodiazepines [14] and z-hypnotics [15]. Already underdiagnosed and undertreated based on the gender discrepancy between diagnoses and suicide rates [16], depression in men worsened during the pandemic [17]. TT has shown a positive, albeit modest, effect on mood and depressive symptoms in older men with hypogonadism [18], so perhaps a similar pattern could explain patients seeking this therapy for symptom alleviation.

The magnitude of change in new patients treated may stem from guideline-discordant care offered by direct-to-consumer platforms [19] given their ease of access. Our prior study performed on data solely from Texas showed that most growth in the TT population came from high-volume prescribers treating over one thousand patients per year, five hundred times greater than the average prescriber, and that the average patient filled two or fewer prescriptions over a 4-year period, not reflecting a long-term treatment [11]. These younger populations receiving more TT have largely not been evaluated in prior research and thus we do not

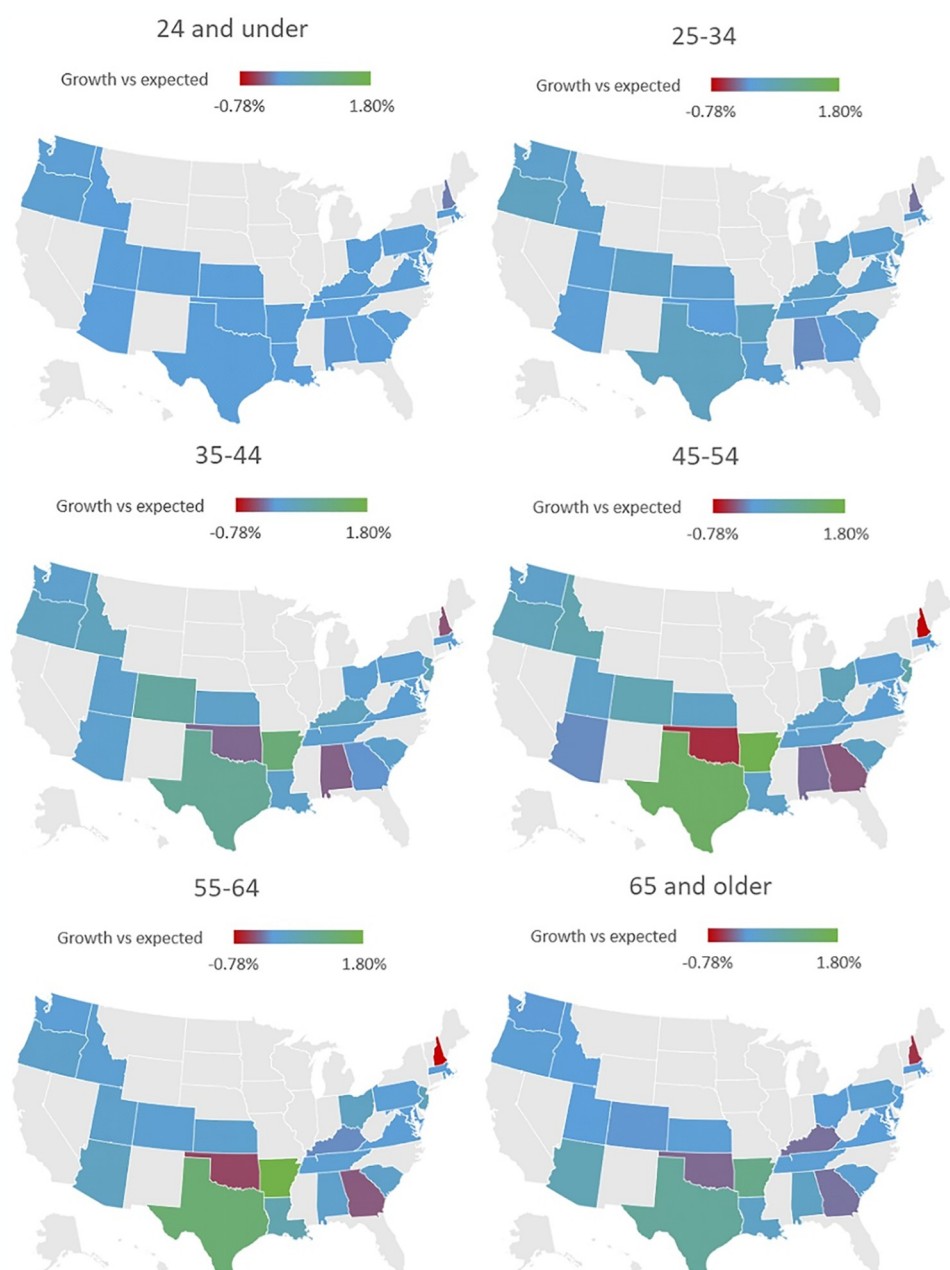

**Fig 3. 2021–22 growth of population receiving TT compared to 2018–2020 forecasting.** Representation of the difference in predicted vs actual prevalence of TT. Prevalence of TT forecasted through 2022 based on prescription data from 2018–2020. Blue states represent difference in prevalence consistent with prior trends, with lower and higher prevalence reflected in gradations of red and green, respectively. Iowa was excluded due to absence of data from 2018.

know the long-term efficacy or adverse effects of therapy initiation or cessation earlier in life. It is unclear why larger increases in prevalence were seen in Texas and Arkansas, but could reflect a combination of physician practice, public interest, and regulatory oversight methods.

Even when TT is clearly indicated, the marginal clinical efficacy is reflected in the low adherence rates seen both in clinical trials [3] and real-world studies [20]. Claims of its ability to modify diabetes risk and control have come under question as well [21]. While TT did not

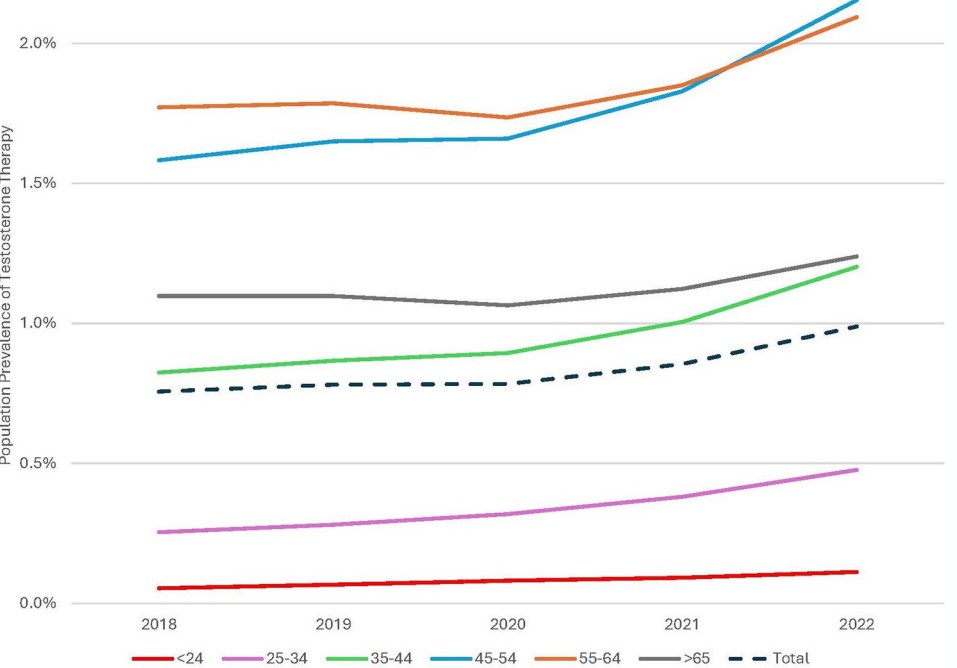

**Fig 4. Population prevalence of testosterone therapy over time.**

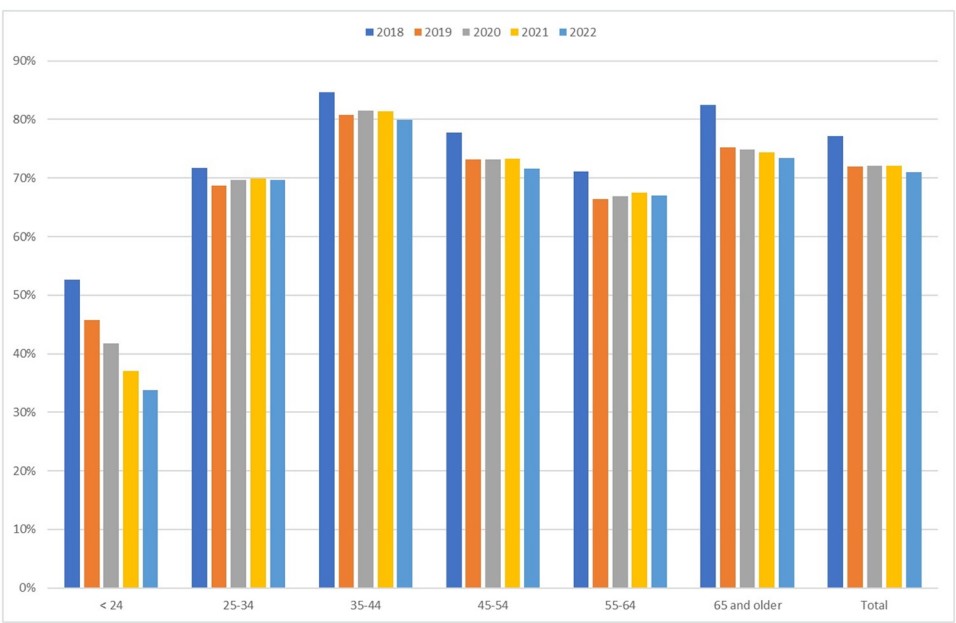

**Fig 5. Percentage of testosterone prescriptions filled by men by age group over time.** Data only available for AZ, AR, IA, KY, NJ.

show an increased risk of major adverse cardiac events (MACE) in a high-risk population [22], most research has been limited to topical gels and not the more widely used injectable formulations that have shown a higher incidence of MACE, hospitalizations, and death [23]. Emerging evidence suggests that middle-aged and older men treated for hypogonadism with TT showed increased risks of concerning arrhythmias, pulmonary embolism, acute kidney injury, and fractures [3, 24], bearing concern for the broader effects on the population from such a sharp uptick in utilization.

Our study's main strength is the utilization of state PDMP databases with mandatory inclusion of all dispensed prescriptions regardless of payor, in contrast to commercial databases and is the first to report new TT prescribing data since 2018. The weaknesses inherent in the study stem from the databases available—lack of identifying information, diagnoses and testosterone levels preclude further analysis of disease prevalence and appropriateness of care. The databases also do not include testosterone formulations administered in a clinic setting, so our results underrepresent the scope of the observed TT prescribing practices. The regional differences are heavily impacted by the inability to include data from half the country and there is a lack of prior PDMP data in many cases to further compare temporal trends. As our study was unfunded, the lack of access to assistance prohibited more in-depth statistical analysis to augment our findings and we would encourage further research in this area.

## Conclusions

Results of this cross-sectional study show that during the COVID-19 pandemic, US national trends in the prescriptions for TT significantly increased, exceeding pre-pandemic trends, particularly in younger male adults. Additional research is needed to differentiate the increases due to unmet need vs overprescribing, to discern the impact of TT prescribing in the states unable to participate in our analysis, and to understand the impact of TT in the younger male population not included in prior studies. We strongly encourage TT prescribing pattern analysis by the more populous states who noted legislative prohibition from inclusion in our study (NY, FL, CA, NC, MI) as well as comparison of the interstate surveillance and enforcement methods for controlled substances as this could impact some of the observed differences.

## Supporting information

**S1 Data.**
(XLSX)

## Acknowledgments

Acknowledgements for specific data sets.

Kentucky: Research reported in this publication was supported, in part, by the Cabinet for Health and Family Services, Office of Inspector General.

Maryland: Aggregate counts were prepared by the Office of Provider Engagement and Regulation, Maryland Department of Health (MDH): Prescription Drug Monitoring Program. The Maryland PDMP data provided is considered preliminary and subject to change, pending finalization of PDMP data by data owners. MDH is not responsible for data analyses, interpretations, conclusions, or references to PDMP data. Contents are solely the responsibility of the authors and do not necessarily represent the official views of MDH.

Pennsylvania: These data were supplied by the Office of Drug Surveillance and Misuse Prevention, Pennsylvania Department of Health, Harrisburg, Pennsylvania. The Pennsylvania

Department of Health specifically disclaims responsibility for any analyses, interpretations, or conclusions.

South Carolina: This information is from the records of the prescription monitoring program, South Carolina Department of Health and Environmental Control. Their authorization to release this information does not imply endorsement of this study or its findings by either the prescription monitoring program or the Department of Health and Environmental Control.

## Author Contributions

**Conceptualization:** Scott Selinger.

**Data curation:** Scott Selinger.

**Formal analysis:** Scott Selinger.

**Investigation:** Scott Selinger.

**Methodology:** Scott Selinger.

**Project administration:** Scott Selinger.

**Resources:** Scott Selinger.

**Software:** Aneesh Thallapureddy.

**Supervision:** Scott Selinger.

**Visualization:** Scott Selinger, Aneesh Thallapureddy.

**Writing – original draft:** Scott Selinger, Aneesh Thallapureddy.

**Writing – review & editing:** Scott Selinger.

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
