## [Decision Letter · Decision Letter 0]

10 Jun 2024

PONE-D-24-13515Cross-Sectional Analysis of National Testosterone Prescribing through Prescription Drug Monitoring Programs, 2018-2022PLOS ONE

Dear Dr. Selinger,

Thank you for submitting your manuscript to PLOS ONE. After careful consideration, we feel that it has merit but does not fully meet PLOS ONE’s publication criteria as it currently stands. Therefore, we invite you to submit a revised version of the manuscript that addresses the points raised during the review process. Please pay particular attention to following areas. 

1. It is essential to clarify whether incidence or prevalence was calculated. 2. The conclusion should be limited to what can be arrived from the available results. 3. I would recommend improving the discussion with special emphasis on comparison with global trends during the studied period and trends in the region before the period covered by this study. 

We look forward to receiving your revised manuscript.

Kind regards,

Appuwawadu Mestri Nipun Lakshitha de Silva, MBBS, MD

Academic Editor

PLOS ONE

3. We note that Figure 3 in your submission contain [map/satellite] images which may be copyrighted. All PLOS content is published under the Creative Commons Attribution License (CC BY 4.0), which means that the manuscript, images, and Supporting Information files will be freely available online, and any third party is permitted to access, download, copy, distribute, and use these materials in any way, even commercially, with proper attribution. For these reasons, we cannot publish previously copyrighted maps or satellite images created using proprietary data, such as Google software (Google Maps, Street View, and Earth). For more information, see our copyright guidelines: http://journals.plos.org/plosone/s/licenses-and-copyright.

1. You may seek permission from the original copyright holder of Figure(s) [#] to publish the content specifically under the CC BY 4.0 license. 

5. Please include your tables as part of your main manuscript and remove the individual files. Please note that supplementary tables (should remain/ be uploaded) as separate "supporting information" files.

Reviewers' comments:

Reviewer's Responses to Questions

**Comments to the Author**

1. Is the manuscript technically sound, and do the data support the conclusions?

Reviewer #1: Partly

Reviewer #2: Partly

2. Has the statistical analysis been performed appropriately and rigorously? 

Reviewer #1: N/A

Reviewer #2: I Don't Know

3. Have the authors made all data underlying the findings in their manuscript fully available?

Reviewer #1: No

Reviewer #2: No

4. Is the manuscript presented in an intelligible fashion and written in standard English?

Reviewer #1: Yes

Reviewer #2: Yes

5. Review Comments to the Author

Reviewer #1: In my opinion, the rationale of this kind of analysis is not very clear. The increase of testosterone use can be concerning and may deserve to be monitored, but the overall prevalence is quite low (also in 2022), with the exception of subjects >=45 years old.

Some comments and suggestions:

1. The study is mainly descriptive. In my opinion, a few statistical analyses (e.g. chi-square for trend; chi square test to compare prevalence by gender) may be helpful.

2. In the methods section you stated that incident rates were calculated, and data were reported in Table 1 and figure 4. This means that it was possible to estimate the number of new TT users? It seems likely to me that subjects reported in table 1 are prevalent, and not incident, cases.

3. I would like to suggest to better clarify how the expected incidence (prevalence?) in 2021-2022 was estimated, and what the values reported in figure 3 represents (what does 1.8% stand for?).

4. It is not very clear to me what is your hypothesis for the increease in TT users during the pandemic period. A mis-diagnosis of hyopogonadism, due to an increase in symptoms associated to a decrease in mental well-being?

5. Geographical differences in the prevalence of TT prescription were observed, but not discussed.

Moreover, it seems to me that the greater than expected rise in users in the 2021-2022 period was limited to very few states (mainly Texas and Arkansas).

Have you any hypothesis about the reasons of these differences?

6. Abstract - results section: The sentence "The number of people treated annually with TT increased 439,659, a 27% rise with 80% of this increase during..." is not very clear. Maybe it could be changed in "In 2022 there was a 27% relative increase of subjects treated with TT (+439,659 cases compared with 2018). The increase was more evident in the pandemic period, with a rise in the prevalence from ... to ..."

Reviewer #2: The authors present a evaluation on the trends in testosterone prescription use in the US. As previously seen, the overall trend is upward with more prescriptions being written/filled. The underlying reasons for this are not investigated by the methodology. I therefore caution the authors on the conclusions presented in the abstract and discussion that any specific link to COVID-19, mental health, or marketing practices. The link to the pandemic should be presented less strongly as the number of confounders is unknown and the methodology do not address or assess this link. For example the trend is going upward throughout the study period and the last two years happen to be during the pandemic.

6. PLOS authors have the option to publish the peer review history of their article (what does this mean?). If published, this will include your full peer review and any attached files.

Reviewer #1: No

Reviewer #2: No

---

## [Author Response · Author response to Decision Letter 0]

11 Jul 2024

I believe I have answered all questions and comments sent back in the revision and attached response to reviewers as well as updated the manuscript to conform to the journals formatting requirements

---

## [Decision Letter · Decision Letter 1]

30 Jul 2024

PONE-D-24-13515R1Cross-Sectional Analysis of National Testosterone Prescribing through Prescription Drug Monitoring Programs, 2018-2022PLOS ONE

Dear Dr. Selinger,

Thank you for submitting your manuscript to PLOS ONE. After careful consideration, we feel that it has merit but does not fully meet PLOS ONE’s publication criteria as it currently stands. Therefore, we invite you to submit a revised version of the manuscript that addresses the points raised during the review process.

Please clarify whether Figure 4 reports incidence rather than prevalence and report the how the incidence was reported in the methods section. 

We look forward to receiving your revised manuscript.

Kind regards,

Appuwawadu Mestri Nipun Lakshitha de Silva, MBBS, MD

Academic Editor

PLOS ONE

Journal Requirements:

Reviewers' comments:

Reviewer's Responses to Questions

**Comments to the Author**

1. If the authors have adequately addressed your comments raised in a previous round of review and you feel that this manuscript is now acceptable for publication, you may indicate that here to bypass the “Comments to the Author” section, enter your conflict of interest statement in the “Confidential to Editor” section, and submit your "Accept" recommendation.

Reviewer #1: (No Response)

2. Is the manuscript technically sound, and do the data support the conclusions?

Reviewer #1: Yes

3. Has the statistical analysis been performed appropriately and rigorously? 

Reviewer #1: N/A

4. Have the authors made all data underlying the findings in their manuscript fully available?

Reviewer #1: No

5. Is the manuscript presented in an intelligible fashion and written in standard English?

Reviewer #1: Yes

6. Review Comments to the Author

Reviewer #1: It seems to me that most of the reviewers' comments and suggestions have been addressed in a satisfactory manner. I still have a revision to suggest:

I'm still not very convinced that in Figure 4 it is reported incidence. The values seems very similar to that reported in table 3 (total column). This means that almost all TT users were "new users"? In any case, I would like to suggest to better explain in the methods section (line 87, page 11) how incident users were defined (subjects with a TT claim in a year and with no TT claims in the previous one?)

7. PLOS authors have the option to publish the peer review history of their article (what does this mean?). If published, this will include your full peer review and any attached files.

Reviewer #1: No

---

## [Author Response · Author response to Decision Letter 1]

30 Jul 2024

Please see my attached letter responding to the comment from reviewer #1

---

## [Editor Report · Decision Letter 2]

7 Aug 2024

Cross-Sectional Analysis of National Testosterone Prescribing through Prescription Drug Monitoring Programs, 2018-2022

PONE-D-24-13515R2

Dear Dr. Selinger,

We’re pleased to inform you that your manuscript has been judged scientifically suitable for publication and will be formally accepted for publication once it meets all outstanding technical requirements.

Kind regards,

Appuwawadu Mestri Nipun Lakshitha de Silva, MBBS, MD

Academic Editor

PLOS ONE
---

## [Editor Report · Acceptance letter]

16 Aug 2024

PONE-D-24-13515R2 

PLOS ONE

Dear Dr. Selinger, 

I'm pleased to inform you that your manuscript has been deemed suitable for publication in PLOS ONE. Congratulations! Your manuscript is now being handed over to our production team.

Kind regards, 

on behalf of

Dr. Appuwawadu Mestri Nipun Lakshitha de Silva 

Academic Editor

PLOS ONE